# Biomass-Derived Adsorbent for Dispersive Solid-Phase Extraction of Cr(III), Fe(III), Co(II) and Ni(II) from Food Samples Prior to ICP-MS Detection

Taghrid S. Alomar [1] ![ID], Mohamed A. Habila [2,*] ![ID], Najla AlMasoud [1,*] ![ID], Zeid A. Alothman [2] ![ID], Mohamed Sheikh [2] and Mustafa Soylak [3,4]





1   Department of Chemistry, College of Science, Princess Nourah Bint Abdulrahman University, Riyadh 11671, Saudi Arabia; tsalomar@pnu.edu.sa
2   Department of Chemistry, College of Science, King Saud University, P.O. Box 2455, Riyadh 11451, Saudi Arabia; zaothman@ksu.edu.sa (Z.A.A.); mshaik75@gmail.com (M.S.)
3   Department of Chemistry, Faculty of Sciences, Erciyes University, Kayseri 38039, Turkey; msoylak@gmail.com
4   Turkish Academy of Sciences (TUBA), Cankaya, Ankara 06100, Turkey
*   Correspondence: mhabila@ksu.edu.sa (M.A.H.); nsalmasoud@pnu.edu.sa (N.A.); Tel.: +966-14-674-198 (M.A.H.); +966-11-8236046 (N.A.); Fax: +966-14-675-992 (M.A.H.)

**Abstract:** A biomass-derived adsorbent was simply prepared and applied as efficient and low-cost solid-phase supports. The adsorbent material was characterized by scanning electron microscopy (SEM), energy dispersive spectroscopy (EDS), X-ray diffraction (XRD), surface area analysis and Fourier transform infrared (FTIR) spectroscopy. The amorphous structure of the prepared adsorbent was indicated from the XRD. The prepared adsorbent exhibited surface functional groups such as carbonyl and hydroxyl groups, which enhance the application of DSPE. An accurate separation and preconcentration of Cr(III), Fe(III), Co(II) and Ni(II) prior to ICP-MS detection was achieved using the biomass-derived adsorbent. The extraction process was performed at pH 4 using 1 mL of 0.5 N nitric acid for elution and recovery of ions. The prepared biomass-derived adsorbent showed efficient performance for extraction application, exhibiting a preconcentration factor of 50 and LODs of 1.4, 2.4, 1.9 and 3.0 $\mu g.L^{-1}$ for Cr(III), Fe(III), Co(II) and Ni(II), respectively, while the LOQs were reported as 4.1, 7.3, 5.7 and 8.9 $\mu g.L^{-1}$ for Cr(III), Fe(III), Co(II) and Ni(II), respectively. The DSPE procedure presented was successfully applied to the determination of the Cr(III), Fe(III), Co(II) and Ni(II) contamination in some food samples.

**Keywords:** biomass-derived adsorbent; preconcentration; adsorption; food samples; heavy metals; ICP-MS

## 1. Introduction

Heavy metals, such as Cr(III), Fe(III), Co(II) and Ni(II), commonly exist in environmental surroundings including soil, water, plants, food and air [1,2]. Numerous health conditions related to inflammation and degenerative diseases and cancer have been associated with exposure to the elements Cr and Co in the workplace environment. The environmental effect of heavy metal emissions is also a huge concern, as they are hard to remove from the ecosystem and cause prolonged negative effects [3,4]. Determining traces of metals in real samples proved to be problematic for two reasons: The first problem is related to concentration levels, as most trace metals exist at levels too low to be detected by analytical instruments. The second is an issue inherent in the buildup or matrix (coexisting ions), as it creates certain barriers to the analytical process. The processes of preconcentration and extraction are possible solutions to these problems [5–7]. Especially, the preconcentration studies are more important in the case of using less sensitive instruments such as inductively coupled plasma optical emission spectroscopy (ICP-OES) and flame/graphite atomic absorption spectroscopy (F/G-AAS). However, in the case of using inductively coupled

plasma mass spectroscopy (ICP-MS), which has a low limit of detection, the extraction procedures serve for sample purification for reducing the matrix effects.

The most common techniques for preconcentration methods are liquid–liquid extraction [1], ion exchange [8], coprecipitation [9], solid-phase extraction [10], dispersive solid-phase extraction (DSPE) and electrochemical deposition [11,12]. Particularly, dispersive solid-phase extraction (DSPE) [13] is one of the most practical substitutes for liquid–liquid extraction in preparing samples due to easy operation and low solvent consumption [14]. In solid-phase extraction, an efficient adsorbent such as Chelex 100 [14], silica spheres [15], polyurethane foam [16] or carbon [17] is applied. For example, Rajabi et al. (2017) applied the layered double hydroxides of aluminum and magnesium with 4-amino-5-hydroxyl-2,7-naphthalenedisulfonic acid monosodium salt interlayer anions for dispersive solid-phase extraction of $Cd^{2+}$, $Cr^{6+}$, $Pb^{2+}$, $Co^{2+}$ and $Ni^{2+}$ prior to detection by flame atomic absorption spectrometry (FAAS) [13]. Feist and Sitko (2019) developed an –extraction process for preconcentration of Pb(II), Cd(II), Zn(II), Cr(III), Mn(II) and Fe(III) using graphene oxide (GO) nanosheets before detection by inductively coupled plasma optical emission spectrometry (ICP-OES) [18].

Biomass is considered as a cheap source of adsorbent materials [19,20]; however, the multistep preparation procedures for biomass activation and conversion to activated carbon consume time, energy and chemicals [21,22]. For example, Anisuzzaman et al. (2015) summarized the preparation of activated carbon from cattail leaves by physical and chemical activation using phosphoric acid in a two-stage process including thermal treatment at 200 °C then activation at 500 °C in muffle furnace [23]. Zhou et al. (2018) investigated the fabrication of activated carbon from waste tea by physical activation using steam with a specific surface area of about 995 $m^2/g$ at 800 °C with a water flow rate of 0.075 g/min and a constant hold time of 0.5 h [24]. Mistar et al. (2020) synthesized activated carbon synthesized from *Bambusa vulgaris* striata with two-step chemical activation and thermal activation at 800 °C [25]. These complicated preparation processes add more cost to the final adsorbent product, resulting in losing the advantage of biomass as a cheap adsorbent. The reason for this additional cost is related to applying a high temperature of around 800 °C for carbonization and/or activation [26–29]. However, in this work, the adsorbent preparation process involved impregnation at room temperature and drying at 105 °C, which saved energy during fabrication procedures. The aims of this research were to apply simple nitric acid treatment at room temperature to prepare a biomass-derived adsorbent and to characterize the adsorbent by scanning electron microscopy (SEM), energy dispersive spectroscopy (EDS), X-ray diffraction (XRD), surface area analysis and Fourier transform infrared (FTIR) spectroscopy to assess the morphology and surface properties. Furthermore, the application of the prepared adsorbent to extraction of Cr(III), Fe(III), Co(II) and Ni(II) from food sample extracts was investigated using dispersive solid-phase extraction (DSPE), and the DSPE process was optimized to achieve efficient and repeatable analytical performance.

## 2. Experiment

### 2.1. Reagents and Materials

The applied chemical and reagents were of analytical-grade purity. Sodium hydroxide, disodium hydrogen phosphate, potassium dihydrogen phosphate, hydrochloric acid, nitric acid and acetic acid were purchased from Merck (Darmstadt, Germany). ICP multi-element standard solution, $Al(NO_3)_3$, $KCl$, $Pb(NO_3)_2$, $Na_3PO_4$, $Cd(NO_3)_2$, $MgCl_2$, $CaCl_2$, $Cu(NO_3)_2$, $KNO_3$, $NaF$, $Na_2CO_3$, $Na_2SO_4$, $NaNO_3$, $Cr(NO_3)_3$, $Fe(NO_3)_3$, $Co(NO_3)_2$ and $Ni(NO_3)_2$ were obtained from Sigma-Aldrich (St. Louis, MO, USA). Phosphate buffer (prepared from disodium hydrogen phosphate and potassium dihydrogen phosphate) was used to adjust the pH of the metal solution during the extraction process. Deionized water was obtained from Milli-Q and used for all solution preparation and dilutions. Inductively coupled plasma mass spectroscopy (ICP-MS) (PerkinElmer, Inc., Shelton, CT, USA) was used for heavy metal determination.

### 2.2. Preparation of the Biomass-Derived Adsorbent

The applied biomass was collected from palm waste of palm tree trunks in Riyadh City, Saudi Arabia. The collected samples were cut into small 1 cm pieces then ground and sieved to between 50 and 100 μm. Then, 5 g of biomass samples were treated with 250 mL of 2 M nitric acid for 10 h, washed with distilled water until the pH was between 6 and 7 and then dried in an oven at 105 °C for 24 h. The washing process was assessed by filtration using Whatman filter paper No. 3. The surface of the prepared biomass-derived adsorbent was characterized with SEM, EDS, BET surface area and FTIR spectroscopy, and the structure was examined using XRD (notes about the applied instruments are available in the Supplementary Materials).

### 2.3. Development of the Dispersive Solid-Phase Extraction onto the Biomass-Derived Adsorbent

The adsorption capacity for Cr(III), Fe(III), Co(II) and Ni(II) of the prepared biomass-derived adsorbent was determined by investigating various masses including 0.01, 0.02, 0.04, 0.06, 0.08 and 0.1 $g.L^{-1}$ mixed with 20 mL of 25 $mg.L^{-1}$ heavy metal ion solution. A water bath was then used to shake the mixture for 1 h at 150 rpm. Then, the phases were separated by centrifugation, and the remaining Cr(III), Fe(III), Co(II) and Ni(II) were analyzed by ICP-MS (the operating conditions are described in Tabls S1). The adsorption capacity was calculated from Equation (1):

$$q_e = \frac{(C_0 - C_e) * V}{M} \tag{1}$$

where $C_0$ represents the initial concentration of metal ions in the solution, $C_e$ is the equilibrium concentration of metal ions in the solution, $V$ is volume of the solution (*L*) and *M* is the mass of the adsorbent (g).

For the dispersive solid-phase extraction optimization, a 50 mL of sample solution including a mixed heavy metal standard of Cr(III), Fe(III), Co(II) and Ni(II) was adjusted to the desired pH by adding 2 mL of phosphate buffer and then adding drops of diluted hydrochloric acid (0.01 M) or sodium hydroxide (0.01 M). The biomass-derived adsorbent was then added with a mass of 0.1 g. A water bath was used to shake the mixtures at 150 rpm for 10 min, and then the biomass-derived adsorbent was separated by centrifugation at 3000 rpm for 4 min. For recovery of adsorbed heavy metal ions from the biomass surface, 1 mL of 0.5 M nitric acid was added and hand-shaken for 1 min, and the phases were separated by centrifugation at 3000 rpm for 4 min. The recovered heavy metal mixture including Cr(III), Fe(III), Co(II) and Ni(II) was analyzed by ICP-MS (the condition for analysis is available in the Supplementary Materials). The recovery was calculated using Equation (2):

$$Recovery\% = \left(\frac{C_f}{C_0}\right) * 100, \tag{2}$$

where $C_f$ is the final concentration and $C_0$ is the initial concentration.

The same procedures were repeated to investigate the rule of pH, eluent type, sample volume, shaking time, heavy metal concentration and coexisting ions. All treatments were applied in triplicate, and blank experiments were conducted in all cases. The intraday and interday precision was estimated by analyzing 0.4 $mg.L^{-1}$ of Cr(III), Fe(III), Co(II) and Ni(II) with 8 replicates at different time intervals over three consecutive days [30].

### 2.4. Real Sample Analysis

The commonly used food samples, including Piper nigrum, Vicia faba, Curcumin, Pimpinella anisum, Cuminum cyminum, cinnamon, Coriandrum sativum, Prunus dulcis, Coriandrum sativum, Lens culinaris and Cicer arietinum, were collected in dry state from Riyadh City markets. One gram of each food sample underwent wet digestion by adding 10 mL of concentrated nitric acid and heat until near dryness. Then, 5 mL of hydrogen peroxide was added, and heating was maintained until a clear extract was obtained, which

was then filtered with Whatman filter paper No. 3, and deionized water was added up to 10 mL [31,32]. Then, 2 mL of each food extract was taken, and the extraction procedure outlined in Section 2.3 was applied to preconcentrate Cr(III), Fe(III), Co(II) and Ni(II) before ICP-MS analysis to enable efficient determination at trace levels. Addition/recovery of food sample extracts was evaluated by adding 0.1, 0.3 and 0.6 mg.L$^{-1}$ from the heavy metal mixture solution of Cr(III), Fe(III), Co(II) and Ni(II), and the extraction procedure outlined in Section 2.3 was repeated to evaluate the recovery and investigate the accuracy and linearity of the developed extraction procedures.

## 3. Results and Discussion

### 3.1. Characteristics of the Biomass-Derived Adsorbent

The prepared biomass-derived adsorbent was examined with SEM and various magnifications of 2000 and 3000 as shown in Figure 1A,B, respectively. The biomass-derived adsorbent has clear pores in the micro size, and the material surface is rich with edges and layers. The EDS analysis (Figure 2) indicates the presence of C and O as the main components, as well as Al, Si, S and P as minor components, which may contaminate the materials because of the waste origin of this biomass-derived adsorbent. The BET surface area for nitrogen adsorption/desorption was determined to be around 4.9462 m$^2$/g. Although the fabricated adsorbent has a relatively low surface area, the presence of groups containing electronegative atoms such as O, S and P enhances the tendency for heavy metal adsorption and enables real application for extraction purposes [33,34].

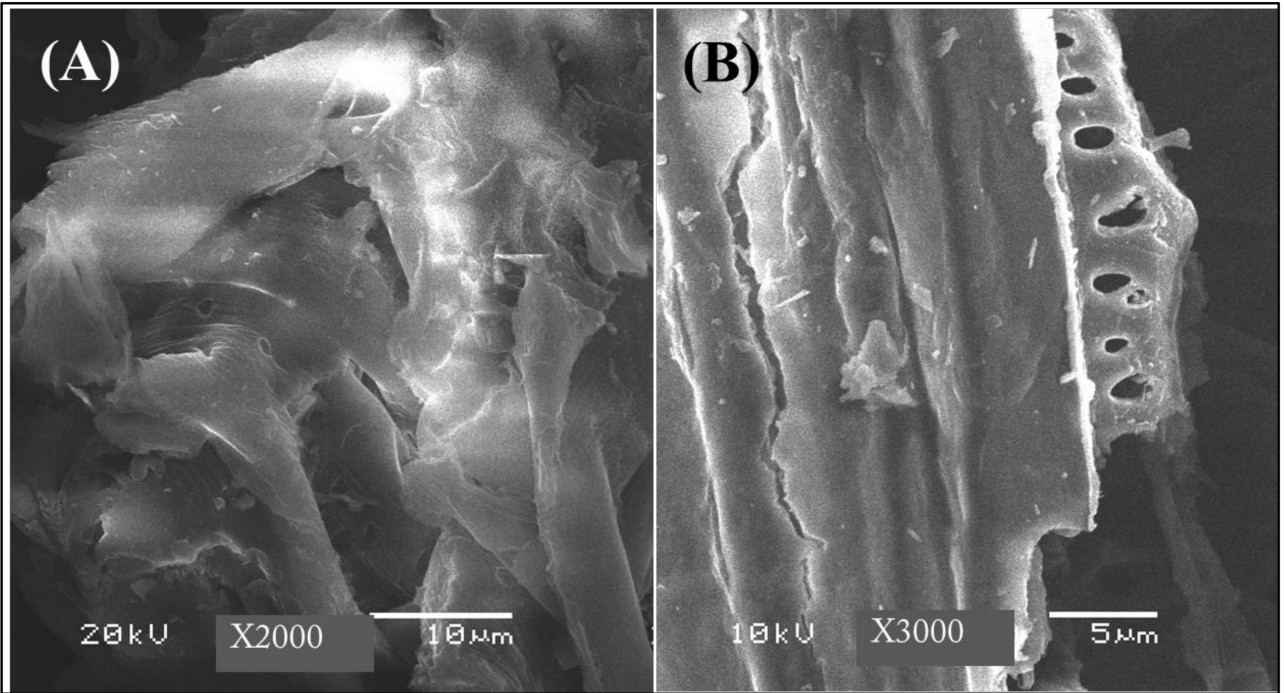

**Figure 1.** SEM images of the biomass-derived adsorbent at various magnifications of 2000 (**A**) and 3000 (**B**).

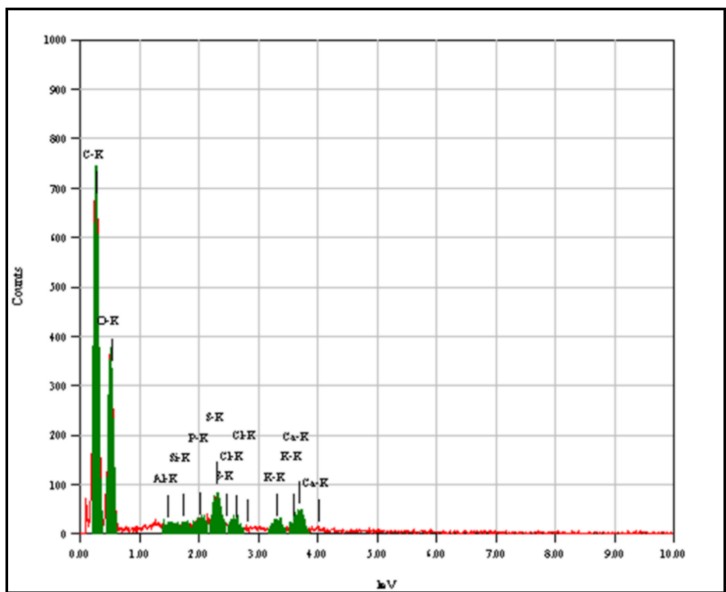

**Figure 2.** EDS of the biomass-derived adsorbent.

The X-ray diffraction to characterize the structural nature of the prepared biomass-derived adsorbent was performed, and peaks are shown in Figure 3. The original palm waste (Figure 3A) and the biomass-derived adsorbent (Figure 3B) showed mainly amorphous structure, with two broad peaks at two theta of 16 and 25 related to the cellulosic content [35]. In addition, the surface functional groups were analyzed with FTIR spectroscopy as shown in Figure 3. The peaks related to OH groups were detected between 3300 and 3500 cm$^{-1}$ in both the original palm waste (Figure 4A) and the biomass-derived adsorbent (Figure 4B). In addition, the aliphatic CH group was detected at around 2900 cm$^{-1}$, C=C appeared at about 1640 cm$^{-1}$ and the carbonyl group appeared at about 1700 cm$^{-1}$. The peaks detected at 1035 and 560 cm$^{-1}$ may be attributed to $PO_4{}^{-3}$ groups [36], which contaminated the palm waste. These surface-active groups are expected to enhance the adsorption [34] process and the application to extraction using adsorption/desorption procedures.

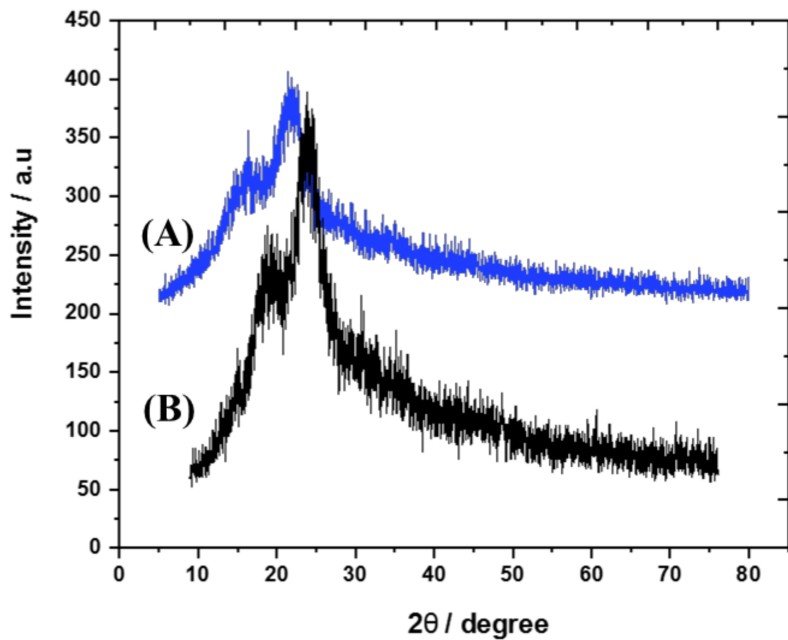

**Figure 3.** XRD of the original palm waste (**A**) and biomass-derived adsorbent (**B**).

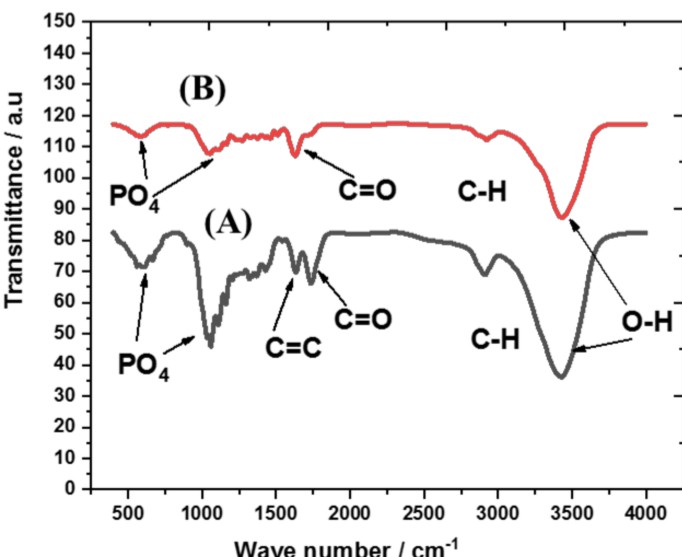

**Figure 4.** FTIR spectrum of the original palm waste (**A**) and biomass-derived adsorbent (**B**).

### 3.2. Development of the Dispersive Solid-Phase Extraction Using the Biomass-Derived Adsorbent

The adsorption capacity of the prepared biomass-derived adsorbent towards uptake of Cr(III), Fe(III), Co(II) and Ni(II) from the aqueous solution of 25 mg.L$^{-1}$ was investigated (Figure 5A). The increasing adsorbent mass led to a decrease in the adsorption capacity, with a maximum adsorption capacity of 51, 50, 51 and 40 mg g$^{-1}$ for Cr(III), Fe(III), Co(II) and Ni(II), respectively, using an adsorbent mass of 0.01 g.L$^{-1}$. The further increase in adsorbent mass from 0.04 to 0.1 g.L$^{-1}$ did not show a noticeable change due to the fixation of the metal ion solution concentration (25 mg.L$^{-1}$) [37]. The adsorbent mass of 0.01 g was selected as the optimal mass in all extraction procedures. The preconcentration experiments were performed using the dispersive solid-phase extraction techniques for enrichment of Cr(III), Fe(III), Co(II) and Ni(II). The developed dispersive solid-phase extraction was optimized by studying the effect of shaking time on the recovery of heavy metals (Figure 5B), indicating that 10 min is enough to achieve a quantitative recovery percentage. The developed biomass-derived adsorbent is rich with surface carbonyl and hydroxyl groups, which enhance the fast uptake of heavy metals.

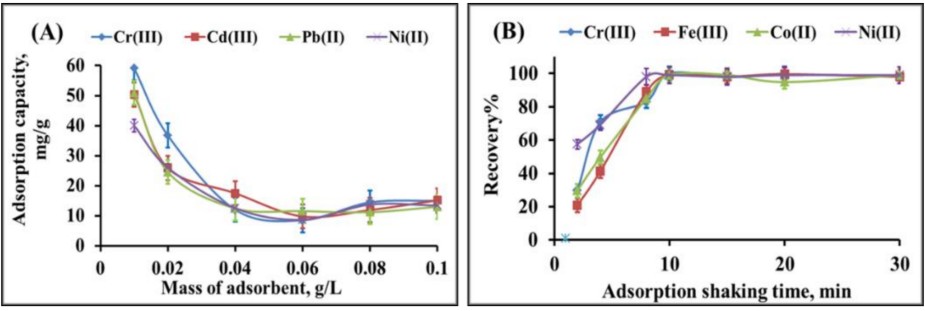

**Figure 5.** Adsorption capacity in relation to mass of adsorbent (**A**) and effect of shaking time on the recovery percentage of Cr(III), Fe(III), Co(II) and Ni(II) during dispersive solid-phase extraction onto biomass-derived adsorbent (**B**).

The developed dispersive solid-phase extraction was further optimized by evaluating the effect of pH of the extraction medium and the possible elution by acids, as well as the ability to tolerate the coexisting ions [38,39]. The effect of pH on the recovery percentage of Cr(III), Fe(III), Co(II) and Ni(II) was studied in the range of 2 to 8; pH 4 exhibited the highest recovery percentage, as presented in Figure 6A. In neutral and basic media, the

recovery percentage decreased, which may be attributed to the weak interaction between heavy metal ions and the biomass in presence of a solution rich in hydroxyl groups.

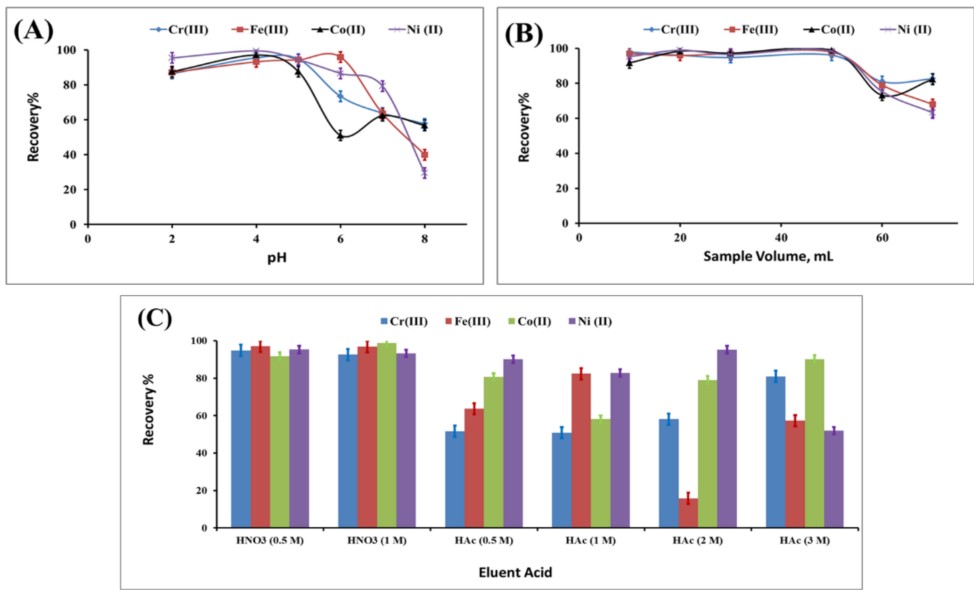

**Figure 6.** Optimization of the dispersive solid-phase extraction using the biomass-derived adsorbent: effect of pH (**A**), effect of sample volume (**B**) and effect of eluent acid (**C**).

The starting sample volume of the metal solution including Cr(III), Fe(III), Co(II) and Ni(II) was changed from 15 to 70 mL during the extraction process, and the related recovery percentage for Cr(III), Fe(III), Co(II) and Ni(II) is presented in Figure 6B. The recovery percentage for Cr(III), Fe(III), Co(II) and Ni(II) was found to be quantitative up to 50 mL. Under constant dose of heavy metal content in the sample, the increase in sample volume led to two effects: the first effect is that the overall metal concentration is diluted in the sample, and the second effect is that the metal ions have to migrate for longer distance to be adsorbed on the biomass solid phase, leading to low recovery percentage.

The efficiency of the eluent acid was studied by reporting the recovery percentage of Cr(III), Fe(III), Co(II) and Ni(II) using 1 mL of 0.5 M nitric acid, 1 M nitric acid, 0.5 M acetic acid, 1 M acetic acid, 2 M acetic acid and 3 M acetic acid (Figure 6C). The elution with 1 mL of 0.5 M nitric acid was the most suitable elution of all the tested metals including Cr(III), Fe(III), Co(II) and Ni(II). The lower recovery using acetic acid as eluent is due to the weak acidity, which exhibited weak elution power for the adsorbed Cr(III), Fe(III), Co(II) and Ni(II), while nitric acid is known as a strong acid and has high dissolving power for elution.

### 3.3. Investigation of the Analytical Performance

The preconcentration factor (PF) was calculated from Equation (3):

$$PF = \frac{Initial\ sample\ volume}{Final\ sample\ volume} \tag{3}$$

The limits of detection (LOD) was calculated using Equation (4):

$$LODs = \frac{(3 * STD)}{m} \tag{4}$$

where STD is the standard deviation of seven blank readings and $m$ is the experimental preconcentration factor, calculated considering the ratio of the slopes of the calibration curves with and without preconcentration procedure.

The performance of biomass-derived adsorbent for extraction application showed efficient recovery percentage for Cr(III), Fe(III), Co(II) and Ni(II), exhibiting a preconcen-

tration factor of 50 and LODs of 1.4, 2.4, 1.9 and 3.0 μg.L$^{-1}$ for Cr(III), Fe(III), Co(II) and Ni(II), respectively, while the LOQs were reported as 4.1, 7.3, 5.7 and 8.9 μg.L$^{-1}$ for Cr(III), Fe(III), Co(II) and Ni(II), respectively. In addition, the optimized dispersive solid-phase extraction process exhibited an RSD% of 4.3, 4.1, 3.2 and 4.9 for Cr(III), Fe(III), Co(II) and Ni(II), respectively, calculated from seven replicates of a metal solution of 0.3 mg.L$^{-1}$. Furthermore, the addition/recovery of food sample extracts was evaluated by adding 0.1, 0.3 and 0.6 mg.L$^{-1}$ from the heavy metal mixture solution of Cr(III), Fe(III), Co(II) and Ni(II) (Table 1). For all tested concentrations of Cr(III), Fe(III), Co(II) and Ni(II), the recovery percentage was in the range of 87 to 100, with major cases between 90 and 100. The linearity of the addition/recovery tests from *Piper nigrum* samples is shown in Figure 7, exhibiting correlation coefficient (R$^2$) values of 99, 98, 1 and 99 for Cr(III), Fe(III), Co(II) and Ni(II), respectively. Moreover, the precision and reproducibility were investigated by studying the intraday and interday precision for Cr(III), Fe(III), Co(II) and Ni(II) from the solution of 0.4 mg.L$^{-1}$ (N = 8) (Table 2). The values of RSD% during intraday and interday investigation were in the range of 2.1 to 8.5 for all tested cases, indicating high accuracy and reproducibility of the developed extraction process.

**Table 1.** Addition/recovery study for the developed extraction of Cr(III), Fe(III), Co(II) and Ni(II) (n = 3).

| Spiking Food Samples | | Cr(III) Found | Recovery % | Fe(III) Found | Recovery % | Co(II) Found | Recovery % | Ni (II) Found | Recovery % |
|---|---|---|---|---|---|---|---|---|---|
| **Food samples** | *Piper nigrum* | 0.02 ± 0.001 | - | 0.37 ± 0.050 | - | 0.01 ± 0.002 | - | 0.08 ± 0.001 | - |
| | *Vicia faba* | 0.09 ± 0.001 | - | BDL | - | BDL | - | 0.09 ± 0.005 | - |
| | *Curcumin* | 0.01 ± 0.000 | - | 0.23 ± 0.009 | - | 0.01 ± 0.001 | - | BDL | - |
| **Spiking 0.1 (mg.L$^{-1}$)** | *Piper nigrum* | 0.200 ± 0.017 | 100 | 0.408 ± 0.027 | 87 | 0.104 ± 0.019 | 97 | 0.169 ± 0.008 | 95 |
| | *Vicia faba* | 0.191 ± 0.016 | 99 | 0.101 ± 0.005 | 99 | 0.104 ± 0.024 | 99 | 0.181 ± 0.042 | 97 |
| | *Curcumin* | 0.197 ± 0.021 | 100 | 0.331 ± 0.026 | 100 | 0.104 ± 0.055 | 99 | 0.100 ± 0.068 | 100 |
| **Spiking 0.3 (mg.L$^{-1}$)** | *Piper nigrum* | 0.389 ± 0.005 | 97 | 0.608 ± 0.047 | 91 | 0.301 ± 0.062 | 98 | 0.366 ± 0.066 | 97 |
| | *Vicia faba* | 0.390 ± 0.013 | 99 | 0.300 ± 0.020 | 99 | 0.302 ± 0.044 | 99 | 0.367 ± 0.057 | 95 |
| | *Curcumin* | 0.390 ± 0.018 | 98 | 0.520 ± 0.064 | 98 | 0.304 ± 0.070 | 100 | 0.300 ± 0.046 | 100 |
| **Spiking 0.6 (mg.L$^{-1}$)** | *Piper nigrum* | 0.690 ± 0.034 | 99 | 0.969 ± 0.041 | 100 | 0.596 ± 0.049 | 98 | 0.668 ± 0.033 | 99 |
| | *Vicia faba* | 0.690 ± 0.048 | 100 | 0.581 ± 0.034 | 96 | 0.594 ± 0.026 | 98 | 0.666 ± 0.070 | 97 |
| | *Curcumin* | 0.691 ± 0.052 | 99 | 0.821 ± 0.030 | 99 | 0.591 ± 0.049 | 98 | 0.597 ± 0.094 | 99 |

The presence of foreign (interfering) ions may occupy the surface-active groups in the biomass-derived adsorbent, leading to a low extraction efficiency due to the competition between ions. Therefore, evaluation of the ability of the developed extraction procedure using the biomass-derived adsorbent was conducted by testing the presence of some common ions including Al$^{+3}$, K$^+$/Cl$^-$, Pb$^{+2}$, PO$_4^{-3}$, Cd$^{+2}$, Mg$^{+2}$, Ca$^{+2}$, Cu$^{+2}$, NO$_3^-$, F$^-$, CO$_3^{-2}$, SO4$^{-2}$ and Na$^+$. The recovery percentage of Cr(III), Fe(III), Co(II) and Ni(II) is presented in Table 3 showing values between 90 and 100, indicating the high tolerance ability of the developed DSPE [15,40].

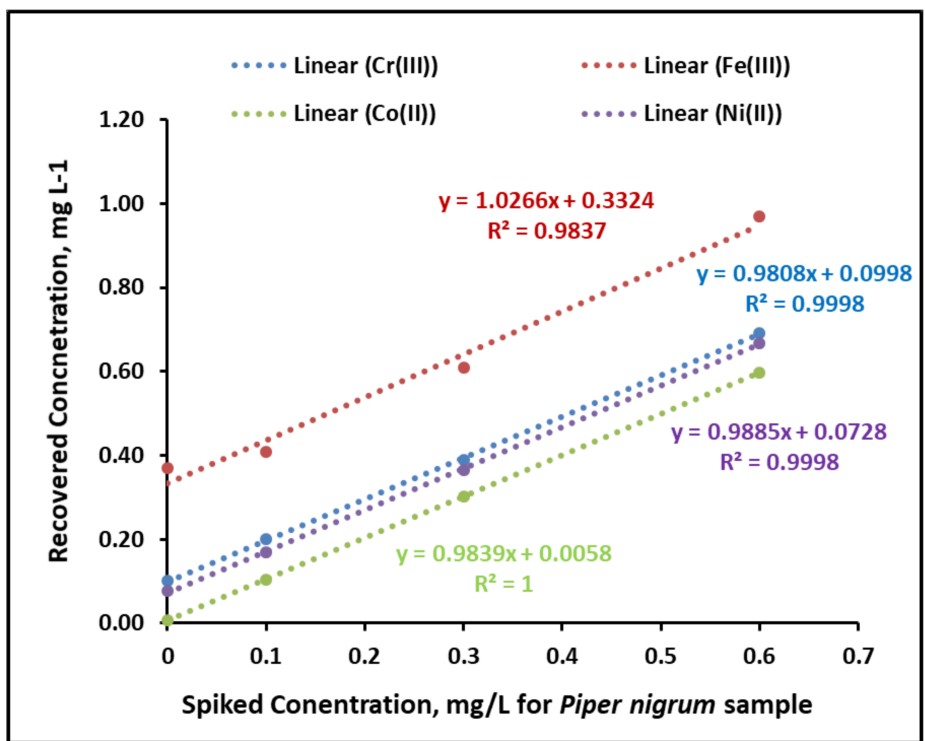

**Figure 7.** The linearity of the addition/recovery tests from *Piper nigrum* samples.

**Table 2.** Evaluation of intraday and interday precision for Cr(III), Fe(III), Co(II) and Ni(II) (N = 8).

| Concentration of Standard Heavy Metal Ion Solution of 0.4 mg.L$^{-1}$ | Intraday Analysis | | | Interday Analysis | | | | | | | | |
|---|---|---|---|---|---|---|---|---|---|---|---|---|
| | | | | First Day | | | Second Day | | | Third Day | | |
| | Detected Concentration (mg.L$^{-1}$) | Accuracy % | Precision (RSD%) | Detected Concentration (mg.L$^{-1}$) | Accuracy % | Precision (RSD%) | Detected Concentration (mg.L$^{-1}$) | Accuracy % | Precision (RSD%) | Detected Concentration (mg.L$^{-1}$) | Accuracy % | Precision (RSD%) |
| **Cr(III)** | 0.395 | 0.094 | 6.0 | 0.397 | 0.656 | 3.7 | 0.401 | 0.250 | 7.0 | 0.400 | 0.000 | 2.1 |
| **Fe(III)** | 0.401 | 0.250 | 4.0 | 0.401 | 0.250 | 7.0 | 0.399 | 0.281 | 5.1 | 0.400 | 0.000 | 3.6 |
| **Co(II)** | 0.398 | 0.469 | 4.1 | 0.394 | 0.750 | 4.9 | 0.398 | 0.625 | 4.2 | 0.396 | 0.250 | 5.7 |
| **Ni(II)** | 0.400 | 0.750 | 8.2 | 0.398 | 0.625 | 8.5 | 0.396 | 0.391 | 4.8 | 0.395 | 0.250 | 7.0 |

**Table 3.** Tolerance for the added foreign ions and calculated recovery percentage for Cr(III), Fe(III), Co(II) and Ni(II).

| Coexisting ion | Added as | Concentration (mg.L$^{-1}$) | Recovery % | | | |
|---|---|---|---|---|---|---|
| | | | Cr(III) | Fe(III) | Co(II) | Ni (II) |
| **Al$^{+3}$** | Al(NO$_3$)$_3$ | 10 | 95 | 99 | 93 | 98 |
| **K$^+$/Cl** | KCl | 2500 | 97 | 99 | 95 | 96 |
| **Pb$^{+2}$** | Pb(NO$_3$)$_2$ | 10 | 96 | 98 | 93 | 93 |
| **PO$_4$$^{3-}$** | Na$_3$PO$_4$ | 500 | 99 | 96 | 94 | 96 |

**Table 3.** *Cont.*

| Coexisting ion | Added as | Concentration (mg.L$^{-1}$) | Recovery % | | | |
|---|---|---|---|---|---|---|
| | | | Cr(III) | Fe(III) | Co(II) | Ni (II) |
| Cd$^{+2}$ | Cd(NO$_3$)$_2$ | 10 | 97 | 98 | 100 | 98 |
| Mg$^{+2}$ | MgCl$_2$ | 200 | 95 | 96 | 96 | 91 |
| Ca$^{+2}$ | CaCl$_2$ | 150 | 94 | 99 | 95 | 93 |
| Cu$^{+2}$ | Cu(NO$_3$)$_2$ | 10 | 96 | 95 | 93 | 98 |
| NO$_3{}^-$ | KNO$_3$ | 1200 | 98 | 96 | 92 | 96 |
| F$^-$ | NaF | 500 | 97 | 95 | 96 | 93 |
| CO$_3{}^{-2}$ | Na$_2$CO$_3$ | 1000 | 97 | 98 | 100 | 98 |
| SO$_4{}^{-2}$ | Na$_2$SO$_4$ | 800 | 100 | 96 | 96 | 91 |
| Na$^+$ | NaNO$_3$ | 12000 | 94 | 99 | 95 | 93 |

### 3.4. Real Food Sample Analysis

The food quality may be affected by metal contamination or bioaccumulation through the food, eventually reaching the human body. Human health is affected by environmental pollution, and contaminated food is considered as a cause of human exposure to metal ions [41]. The developed solid-phase-based extraction was applied for preconcentration of Cr(III), Fe(III), Co(II) and Ni(II) in some food samples including *Piper nigrum*, *Vicia faba*, *Curcumin*, *Pimpinella anisum*, *Cuminum cyminum*, cinnamon, *Coriandrum sativum*, *Prunus dulcis*, *Coriandrum sativum*, *Lens culinaris* and *Cicer arietinum*. Table 4 shows the concentrations of Cr(III), Fe(III), Co(II) and Ni(II) in the tested real food samples (n = 3), indicating the trace levels within the safe region stated by international organizations related to food quality [42,43]. The comparison of the limits of detection between the developed extraction processes using biomass-derived adsorbent and other methods from the literature [14,18,19,42,43] is presented in Table 5. For Cr(III) and Co(II) extraction, the present work exhibits LODs lower than dispersive solid-phase extraction (USE-AA-D-SPE) [14], while for Fe(III), the LODs were higher than those reported by [19,42], and for Ni(II), they were higher than those reported by [43]. This confirms that the developed biomass-derived adsorbent prepared in this work has a similar performance to graphene- and carbon-nanotube-based materials for preconcentration purposes.

**Table 4.** Application of the proposed extraction for preconcentration of Cr(III), Fe(III), Co(II) and Ni(II) from real food samples (n = 3).

| Real Food Samples | Cr(III) (mg kg$^{-1}$) | Fe(III) (mg kg$^{-1}$) | Co(II) (mg kg$^{-1}$) | Ni (II) (mg kg$^{-1}$) |
|---|---|---|---|---|
| *Pimpinella anisum* | 0.08 ± 0.001 | 0.68 ± 0.015 | 0.11 ± 0.010 | 0.09 ± 0.005 |
| *Cuminum cyminum* | 0.009 ± 0.002 | 0.82 ± 0.023 | 0.01 ± 0.009 | 0.05 ± 0.003 |
| *Cinnamon* | 0.03 ± 0.009 | 1.14 ± 0.026 | 0.09 ± 0.005 | 0.06 ± 0.001 |
| *Coriandrum sativum* | 0.002 ± 0.000 | 2.17 ± 0.180 | 0.12 ± 0.004 | 0.09 ± 0.005 |
| *Prunus dulcis* | 0.079 ± 0.005 | 2.26 ± 0.085 | 0.13 ± 0.007 | 0.07 ± 0.001 |
| *Coriandrum sativum* | 0.012 ± 0.006 | 3.45 ± 0.240 | 0.09 ± 0.004 | 0.05 ± 0.002 |
| *Lens culinaris* | 0.032 ± 0.004 | 1.02 ± 0.09 | 0.07 ± 0.005 | 0.09 ± 0.001 |
| *Cicer arietinum* | 0.076 ± 0.005 | 1.83 ± 0.047 | 0.01 ± 0.001 | 0.07 ± 0.003 |

**Table 5.** Comparison of extraction of Cr(III), Fe(III), Co(II) and Ni(II) using various methods from literature.

| Extraction Process | Limits of Detection ($\mu g.L^{-1}$) | | | | |
|---|---|---|---|---|---|
| | Cr(III) | Fe(III) | Co(II) | Ni (II) | Reference |
| Solid-phase extraction of metal ions from fuel ethanol with a nanostructured adsorbent | - | 0.2 | - | 0.46 | [44] |
| Dispersive solid-phase extraction (USE-AA-D-SPE) Layered double hydroxides with 4-amino-5-hydroxyl-2,7-naphthalenedisulfonic acid monosodium salt interlayer anions | 1.7 | - | 2.1 | 2.4 | [13] |
| Dispersive micro solid-phase extraction (DMSPE) onto graphene oxide (GO) nanosheets | 0.06 | 0.21 | - | - | [18] |
| Magnetic solid-phase extraction using carbon-coated Fe3O4 nanoparticles | 0.002 | - | 0.001 | - | [17] |
| Solid-phase extraction using multiwalled carbon nanotubes impregnated with 4-(2-thiazolylazo)resorcinol | - | - | - | 4.3 | [45] |
| Dispersive solid-phase extraction onto biomass-derived adsorbent | 1.4 | 2.4 | 1.9 | 3.0 | Present work |

## 4. Conclusions

The biomass-derived adsorbent was successfully prepared by nitric acid impregnation and showed a high tendency for adsorption of heavy metals with a capacity of 51, 50, 51 and 40 mg g$^{-1}$ for Cr(III), Fe(III), Co(II) and Ni(II), respectively. In addition, the prepared adsorbent exhibited a superior efficiency for dispersive solid-phase extraction, exhibiting.LODs of 1.4, 2.4, 1.9 and 3.0 $\mu g.L^{-1}$ for Cr(III), Fe(III), Co(II) and Ni(II), respectively. The LOQs were reported as 4.1, 7.3, 5.7 and 8.9 $\mu g.L^{-1}$ for Cr(III), Fe(III), Co(II) and Ni(II), respectively. In addition, the optimized dispersive solid-phase extraction process exhibited RSD% of 4.3, 4.1, 3.2 and 4.9 for Cr(III), Fe(III), Co(II) and Ni(II), respectively. The proposed extraction procedures have a preconcentration factor of 50. In addition, the extraction method using biomass-derived adsorbent showed a high tolerance ability for various ions including $Al^{+3}$, $K^+$/$Cl^-$, $Pb^{+2}$, $PO_4^{-3}$, $Cd^{+2}$, $Mg^{+2}$, $Ca^{+2}$, $Cu^{+2}$, $NO_3^-$, $F^-$, $CO_3^{-2}$, $SO4^{-2}$ and $Na^+$, indicating the accuracy of the developed dispersive solid-phase extraction procedures.

**Supplementary Materials:** The following are available online at https://www.mdpi.com/article/10.3390/app11177792/s1, Table S1: ICP-MS operating conditions.

**Author Contributions:** Data curation, N.A.; formal analysis, T.S.A., N.A. and M.S. (Mohamed Sheikh); funding acquisition, N.A.; investigation, M.A.H., Z.A.A. and M.S. (Mustafa Soylak); methodology, T.S.A., M.A.H. and M.S. (Mohamed Sheikh); project administration, M.A.H., Z.A.A. and M.S. (Mohamed Sheikh); resources, N.A. and Z.A.A.; supervision, Z.A.A.; validation, M.A.H.; writing—original draft, T.S.A.; writing—review and editing, M.A.H. and M.S. (Mustafa Soylak). All authors have read and agreed to the published version of the manuscript.

**Funding:** This work was funded by the Deanship of Scientific Research at Princess Nourah Bint Abdulrahman University through the research group program grant number RGP-1440-0023.

**Institutional Review Board Statement:** Not applicable.

**Data Availability Statement:** The data presented in this study are available on request from the corresponding author. The data are not publicly available due to privacy.

**Acknowledgments:** The authors acknowledge the Deanship of Scientific Research at Princess Nourah Bint Abdulrahman University for funding this research through the research group program grant number RGP-1440-0023.

**Conflicts of Interest:** The authors declare no conflict of interest.

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
