# Peer review of "Biomass-Derived Adsorbent for Dispersive Solid-Phase Extraction of Cr(III), Fe(III), Co(II) and Ni(II) from Food Samples Prior to ICP-MS Detection"

_applsci, doi:10.3390/app11177792_

Round 1

Reviewer 1 Report

This work is eventually publishable, but authors need to address two main concerns associated with this work. First is the introduction. This work describes the determination of heavy metals in food, so authors do not need to provide the research on drawbacks of using heavy metals in implantable devices or other topics discussed in the first paragraph of the introduction which is unrelated to this work. Secondly, the authors described the difference of their work compared with the existing methods to prepare sorbets from biomass and mentioned that their method is faster and does not need heating while they are using acid. Is there any disadvantages of preparing sorbent from biomass regarding using such corrosive regent (nitric acid). This should be discussed.

Specific comments:

Abstract:

Line 14: “…for dispersive solid-phase extraction (DSPE) procedures…”DSPE for what purposes. It should be mentioned here.

Line 25: Please change “dispersive solid-phase extraction” to “DSPE”.

Figure 5 a what was the optimal mass of the sorbent. It should e mentioned in the discussion in lines 196-202.
lines 221-222: please mention the functional groups of the sorbent for better explanation.

In results and discussion (method optimization) authors need to mention what is the optimum condition! “Under constant dose of heavy metal content in the sample, the increase in sample volume led to a decrease in the overall metal concentration and reduced the adsorptive uptake, leading to a low recovery%” Please modify.

Line 248: it is understandable that the recovery is 100% in this work but the correct formula for PF should include the percent recovery. Please revise.

Table 2: please add the accuracy (%) to this table.

Author Response

Thanks for reviewing our manuscript, all comments are considered and the manuscript is improved. Details about the point by point response are submitted in separate file named response for reviewers comments. 

Regards

Reviewer 2 Report

The new version of the manuscript now entitled “Biomass derived adsorbent for dispersive solid-phase extraction of Cr(III), Fe(III), Co(II) and Ni(II) from food samples prior to ICP-MS detection”, has been much improved compared to the first version submitted. The introduction is much better. The authors also added more description of the methods. Nonetheless, several information important for the reproducibility of the method are still missing. In addition, the calculation of the preconcentration factor and LODs are not correct and there are probably mistakes in the interpretation or plotting of the IR-spectra. Therefore, I recommend major revisions. Please find below my detailed comments.

Introduction:

Lines 36-37: Which study is meant? This sentence does not apply to the study described in this manuscript.

Lines 50-51: This sentence is inaccurate, first it would be important to give a range of order of magnitude for the expected concentrations in food samples (since it is the focus of the paper). This would help to determine target LODs for the method which have to be mentioned in the introduction. Second the detection limits of modern ICP-MS instruments are in the ng/L range for Co, Ni, and Cr and in the µg/L range for Fe. This LODs are much below the concentrations of these elements in environmental samples and below the limits proposed in guidelines. Therefore, pre-concentration would make sense mainly for being able to use less sensitive instruments such as ICP-OES, F/G-AAS, etc. Please rephrase and complete.

Lines 50-52: please explain with more details the problems with the matrix in food samples. Is it mainly about isobaric interferences or are they also matrix effects independently from these interferences?

Line 62: what is a “layered double hydroxide”?

Line 63: upper scripts for the charges

Methods:

The description of the methods is still incomplete. The method should be completely reproducible from the description. The following elements are missing and have to be completed:

- ICP-MS model and corresponding equipments (nebuliser type, use of collision cell, etc). In addition, information about tuning, calibration, and internal standard should be given.

- Isotopes used for measuring the elements.

- Description of instruments for SEM, XRD, BET, and IR and the corresponding methods including sample preparation

- Description of the source of biomass (specie(s) of the tree? What “waste” means in this context? Are random pieces of the trunks cut off or there are some industrial processes using the trees and the waste of this process is collected?)

- Volume of nitric acid solution used per g of biomass

- Description of the method for washing the biomass after the acid treatment (filtration? centrifugation?)

- Concentrations used for phosphate buffer

- Filters materials, suppliers, and size cut-off used for the food sample preparation

- Which volume of nitric acid solution was used for re-dissolving the analytes for the food samples? 1 mL as for the method optimization?

- Centrifugation procedure: model of the centrifuge, g-force used for separation + was the supernatant completely removed or an aliquot of a given volume was sampled from the supernatant?

- Spiking procedure

Line 100: lower scripts for nickel nitrate

Line 121: remove “mixed” and add “solution” at the end of the sentence.

Line 122: what does it mean to shake with a water bath? Please explain more clearly. E.g. we used magnetic stirrers or we used

Line 124: the abbreviation ICP-MS was already defined sooner in the tex. Use the abbreviation only.

Line 144: correct for “role”

Line 147: the solutions were in pure water? Please clarify in the text. Did these measurements included the pre-concentration step or it was just the ICP-MS measurements?

Line 160: what is meant by “deamination”? Why are amines suddenly a problem? H2O2 would mainly mineralize organic matter, regardless of the N-content…

In addition, I would be important to check if there is some reaction between the H2O2 and the adsorbent. IR-spectra should be enough to evaluate it.

Results and discussion:

Lines 171-172: the sentence is inaccurate; it is rather the functional groups: carbonate, phosphate, and sulfate which will be responsible for most of the sorption of cations not the elements themselves.

Figure 2: asix labels are too small

Line 181: a reference is needed for the attribution of the amorphous phase.

XRD: Please add an interpretation for the two “peaks” observed in the diffractogram. Furthermore, what could be the reason for the shift observed for the peak positions when comparing A and B?

IR: Similar question, there is a shift between A and B or can it be that one graph was shifted compared to the other when the figure was made? In this case please correct for it. Considering this shift, the peak attributed to C=O in the spectrum B is probably wrong and the C=O should be rather represented in the shoulder right to this peak. The interpretation of the spectrum would then be completely different.

Lines 220-222: could the presence of the phosphate buffer influence the recovery? It is quite unexpected that the sorption of cations on surfaces with more or less negatively charged groups such as carboxylate, phenol, sulfate, and phosphate decreases with pH. The more negative the surface, the higher the sorption of cation should be as long as sorption sites are not saturated.

Figure 6: axis labels are too small.

Definition of PF and LOD: I recommend to re-evaluate the definition of pre-concentration factor and LOD in this study. I copied some doi-links of papers in which the pre-concentration factor is explicitely defined:

https://doi.org/10.1016/j.microc.2004.02.006

http://dx.doi.org/10.1080/03067310903020334

https://doi.org/10.1016/j.chroma.2015.02.068

In all cases the pre-concentration factor is defined as the ratio between the slope of the calibration curve obtain with and without the pre-concentration step, respectively. Therefore, it is higher when the same amount of analyte is concentrated in a smaller volume after the step. The definition proposed in the manuscript results in the opposite situation: the smaller the final volume, the higher the smaller the PF, which contra-intuitive.

Furthermore, if the standard deviation of the ICP-MS signal for the blank is chosen for the LOD’s calculation, then the LOD should be 3*STD divided by the slope of the calibration curve. Otherwise the units does not match. The pre-concentration factor should not be used in the calculation of the LOD from the calibration curve since the effect of the pre-concentration will be accounted in an increase of the calibration slope.

Standard addition: to evaluate if matrix effects are present with and without the pre-concentration steps, the spiked samples should be measured without the extraction steps and the slopes of the curve compared (t-test or ANCOVA) with ionic standards in pure water. The same should be carried out with samples with the extraction step. This is a safe way to prove that the matrix effects can be reduced using the proposed method. The standard addition procedure requires the blank corrected intensity signal as y-axis and the spiked concentration as x-axis. Please refer to the corresponding literature.

Table 3: how were the concentrations of the interfering ions selected?

Line 305: Add the limit values from “international organisations” in the table 4 for comparison and provide a reference for these values.

Table 5: add the unit for the LOD

Author Response

(The authors gave the same response as above.)

Round 2

Reviewer 2 Report

I appreciate the improvements carried out by the authors. Unfortunately, I am sorry to say that there are still a couple of major corrections required. Indeed, the formula 3 and 4 are still wrong despite my remarks and some more information are needed for other researchers to be able to reproduce the method (see below for details). This is a method paper, so the method's description is essential and has to be flawless.

Methods:

The description of the methods is still incomplete. The method should be completely reproducible from the description. The following elements are missing and have to be completed:

- Please add the nebuliser type (e.g. Micromist, Meinhard) and if you use the collision cell or not for the ICP-MS measurements, if yes specify which isotopes were measured with the collision cell and the used collision gases. It’s an important information since the collision cell reduces a lot the polyatomic interferences. In addition, information about tuning (daily? Weekly?), calibration (which standards were used? Suppliers? Were the standards matrix-matched or diluted in pure water?), and internal standard (did you use an internal standard? If yes which one?) are required for reproducing the method in foreign labs.

- Isotopes used for measuring the elements using ICP-MS. Hg for example has four isotopes which can be measured with ICP-MS. Each isotope has its own interference pattern and abundance, therefore, the LOD depends on the chosen isotope. The results presented in this study are not comparable without this information.

Results and discussion:

Equation 2: correct the numbering.

IR: I understand the authors’ response and I agree that oxidation through nitric acid treatment could induce a shift in the carbonyl band but it is much less probable that all bands including phosphates and CH are shifted in the same direction with the same shift value. Therefore, please double check that there was no unintentional shift included in the figure, e.g. during the preparation of the figure. Also the fact that the spectra don’t start at the same wavelength suggests that such an error occurred here.

Definition of PF: I agree with the authors’ response BUT the formula they give in their response differs from the formula they have in their manuscript (Vfinal/Vinitial vs Vbefore/Vafter in the response). Please correct it in the manuscript.

Definition of LOD: my previous comment was disregarded. Please correct the formula or the symbols’ definition. Something is obviously wrong with different units left and right of the = sign (µg/L vs cps).

Author Response

The manuscript is revised according to comments as follow:

Comment

I appreciate the improvements carried out by the authors. Unfortunately, I am sorry to say that there are still a couple of major corrections required. Indeed, the formula 3 and 4 are still wrong despite my remarks and some more information are needed for other researchers to be able to reproduce the method (see below for details). This is a method paper, so the method's description is essential and has to be flawless.

Response

Thanks for reviewing our manuscript. We appreciate your efforts and time, and respond to your valuable comments which help us for improving the manuscript.

Comment

Methods:

The description of the methods is still incomplete. The method should be completely reproducible from the description. The following elements are missing and have to be completed:

- Please add the nebuliser type (e.g. Micromist, Meinhard) and if you use the collision cell or not for the ICP-MS measurements, if yes specify which isotopes were measured with the collision cell and the used collision gases. It’s an important information since the collision cell reduces a lot the polyatomic interferences. 

Response

The nebulizer type is Meinhard, and we did not use collision cell.

Comment

In addition, information about tuning (daily? Weekly?), calibration (which standards were used? Suppliers? Were the standards matrix-matched or diluted in pure water?), and internal standard (did you use an internal standard? If yes which one?) are required for reproducing the method in foreign labs.

Response

For calibration, we applied the ICP multi-element standard solution which provided from sigma. The tuning optimization is applied monthly with maintenance company help. In addition, we did not operate internal standard method. However, we applied addition/recovery optimization from food samples matrix.

Comment

- Isotopes used for measuring the elements using ICP-MS. Hg for example has four isotopes which can be measured with ICP-MS. Each isotope has its own interference pattern and abundance, therefore, the LOD depends on the chosen isotope. The results presented in this study are not comparable without this information.

Response

I agree with you, but we did not apply mercury in our study.

Comment

Results and discussion:

Equation 2: correct the numbering.

Response

The number is corrected.

Comment

IR: I understand the authors’ response and I agree that oxidation through nitric acid treatment could induce a shift in the carbonyl band but it is much less probable that all bands including phosphates and CH are shifted in the same direction with the same shift value. Therefore, please double check that there was no unintentional shift included in the figure, e.g. during the preparation of the figure. Also the fact that the spectra don’t start at the same wavelength suggests that such an error occurred here.

Response

The IR Figure is reestablished and corrected.

Comment

Definition of PF: I agree with the authors’ response BUT the formula they give in their response differs from the formula they have in their manuscript (Vfinal/Vinitial vs Vbefore/Vafter in the response). Please correct it in the manuscript.

Response

The equation is corrected in the manuscript.

Comment

Definition of LOD: my previous comment was disregarded. Please correct the formula or the symbols’ definition.

Response

The equation of LOD is corrected in the manuscript.

Comment

Something is obviously wrong with different units left and right of the = sign (µg/L vs cps).

Response

The concentration units is revised and corrected.

This manuscript is a resubmission of an earlier submission. The following is a list of the peer review reports and author responses from that submission.

Round 1

Reviewer 1 Report

This paper describes a biomass sorbent used to extract ions from food samples, followed by analysis using ICP-MS. The authors performed characterization using SEM FT-IR, EDX, and XRD, method optimization in deionized water, and method validation in dried food samples. Although the purpose of the work was exciting, the presented work is not suitable for publication for several significant issues:

1- The introduction is poorly written in fragmented paragraphs without cohesion and theme. Readers must be able to understand the opening, then experiments and results. I recommend this section for a complete English editing.

2-This work is based on extraction using dispersive solid phase extraction (DSPE) and not solid phase microextraction (SPME). The authors must perform a useful literature review on the available microextraction/extraction techniques to provide a suitable introduction to this work.

3- The authors were unable to show the advantages of this work. They mentioned that using biomass as sorbent could be expensive due to complicated processing. But did not note that what is advantageous about their method of preparation that makes this work publishable?

4- The experimental section is not very informative, and lots of details regarding techniques used (extraction or analysis) are missing. Each figure must be informative enough to understand each experiment's conditions, like figure 5 that has no detail. And some further characterization is required, like adsorption to verify the porosity. SEM is useful for morphology but not for porosity.

5- The result and discussion section is also poorly written without a comprehensive explanation. It seems that it has been left to the readers to interpret the results (like optimization of elution solvent).

6- For method validation, the authors only mentioned some LOD and LOQ values but not a full validation like linear range, linearity (R2), accuracy, and precision.

Considering all the shortcomings, I would highly suggest authors re-write the paper, perform some additional experiments, particularly for method validation, including details of the experiments and improving the quality of discussions.

Detailed comments:

The tittle must be changed. The extraction technique that the authors used in this work is dispersive solid phase extraction (DSPE) and not solid phase microextraction (SPME).

Line 15: “……SEM, EDS, XRD and FTIR……”. If authors intend to use the abbreviations, please introduce them first.

Line 22: please modify the superscript in "…μg L-1…"

Please add a comma after Ni(II) in "Ni(II) respectively".

Line 30 Please introduce Cr and Co

Lines 53-54- What does this sentence mean? "Such effects are even found by chemically analyzing under-detectable concentration levels of heavy metals [7]." How it is undetectable, and effects are linked to the presence of heavy metals. If authors meant to consider a very low concentration, please state that and revise in a quantitative manner.

Line 82 "extracting organic compounds and eliminating matrix interferences" the purpose of this work is not to extract organics. Please revise.

Lines 108-109: please modify subscript in chemical formulations.

Section 2. experimental: please provide details of the instrument and operating conditions. No information on ICP-MS is provided.

Please provide the details of other instruments used for characterization like SEM, EDS, FTIR, and XRD.

Line 110: please revise "deionized waste" to "deionized water".

Section 2.1. Reagents and materials: please provide the supplier of the chemicals and moreover, elase revise the preparation of biomass sorbent in a separated section by explaining further details of the procedure like any particle size etc.,

Section 2.2.

  • The tittle is confusing and does not reflect the section. Please revise the tittle of this section from "Development of the solid phase microextraction onto the biomass derived adsorbent" to "microextraction procedure".
  • This section must contain all optimized conditions in a way that reader does not need to see the whole paper to repeat this work. For example, the optimal mass of the sorbent, "mixed with a certain 118 weight of the biomass derived adsorbent". There is no other explanation other than the mentioned sentence in the paper to show how much adsorbent was used?
  • What kind of shaking was used? Please state if magnetic, orbital, vortex shakers were used.
  • How was the sorbent collected? Centrifugation of filtration?
  • What phosphate buffer was used? Please provide details.

Line 132: please change "deamination" to "determination".

Lines 138-139: "The biomass derived adsorbent has clear pores in the micro size" the SEM micrographs are not informative. The porosity must be determined thorough adsorption studies like N2 adsorption followed by BET analysis.

Line 152: please revise the subscript in "3500 cm-1".

Lines 155-156: "These surface active groups are expected to enhance adsorption process and the application for microextraction by adsorption/desorption procedures." Please explain how the adsorption can be improved by such functionalities in the adsorbent?

Line 175: please explain why "The elution with 1 mL of 0.5 M nitric acid was the most suitable elution"? and why nitric acid is a better eluting solvent that acetic acid?

Line 185-186: please delete the fragmented sentence in "In addition, The relative standard deviation (RSD%) was found to be".

Please provide the units in Table 1 for found concentration and recovery and also please use the same unit as discussed in the text. For example. Please revise the unit for added concentration from mg/L to µg/L.

Please explain how LODs and LOQs were Calculated. (i.e., Method blank, instrument blank)

Line 186- What are the concentrations for those reported RSD values?

Please provide further details of the methods validation like R2 values, linear range, accuracy and precision at low, mid and high concentrations of the linear range, Inter-day validation,

Line 195: what does "foreign ions" mean?

Please change the "low microextraction efficiency" to "low extraction efficiency". The microextraction refers to the miniaturized sample size, adsorbent size, apparatus required. While here the intend should be to mention the process.

Table 3. please use the same units throughout the manuscript.

214-215: "indicating the very trace levels which are in the safe region stated by international organization." What is the safe region for these ions in food samples? Pease report them based of mg/per gam (microgram/gram) because the original food samples were dried samples.

References : too many references for such a short paper!

Reviewer 2 Report

In the manuscript entitled “Biomass derived adsorbent for Solid Phase Microextraction of Cr(III), Fe(III), Co(II) and Ni(II) from food samples prior to ICP-MS detection”, a method for producing an adsorbent from biological waste and its application for the preconcentration of four transition metal cations prior to ICP-MS is described.

Developing new analyte enrichment techniques is important to improve the LODs of standard detection techniques. However, it is not clear from the manuscript what is the advantage of the proposed method because the concentrations used for the study were high enough to be determined with standard ICP-MS instruments without preconcentration. If preconcentration has advantages, it will always be preferable to measure samples directly, if the sensitivity allows it. The removal of interferences can be important but the improvement brought by the proposed preconcentration method was not evaluated in this respect. Indeed, matrix effects are mentioned but not evaluated in this study. Furthermore, many essential information about the methods is missing, so that the method description does not ensure reproducibility. Therefore, I recommend rejection of this manuscript. Please find below detailed comments and recommendations.

Introduction:

The structure of the introduction is quite confusing because it goes in different directions (health, environment, food, method development). A clearer more linear structure would help the reader understanding the main aspect of the analytical challenges the authors try to tackle. In general the introduction is quite superficial.

Lines 63-66: the LODs for standard ICP-MS equipment for transition metals is in the high ppt range which is below the natural background of many inorganic pollutants. Therefore, it is unclear from the introduction why the low concentrations are still a challenge and what would be the required LODs.

Lines 66-69: interferences maybe a serious problem for environmental samples and SPE is an approach to tackle it. However, the authors do not provide examples of interferences relevant to the method they applied. Only seawater is mentioned and was not evaluated in the study.

Line 81: SPE = Solid phase extraction, remove “liquids”.

Line 99: since the main of the study was to develop a cheap adsorbent, the price, reagents, and time required for the production should be listed and compared with the currently most used adsorbents for transition metals. Otherwise, it is not clear why this particular adsorbent is competitive and, hence, important to test.

In addition, the authors may consider discussing the option of passive sampling, which is also highly promising for very low concentrations and reduction of matrix effects.

Methods:

The description of the methods is very incomplete. Some examples of what is missing:

- Description of instruments (SEM, XRD, IR, ICP-MS) and measurement methods

- Suppliers for chemicals

- Homogeneization steps for the biomass (grinding?)

- Purity level of water

- Filters materials

- Separation of the biomass from the sample

- Equations/references for recovery and LOD/Q calculation

- Spiking procedure

- Number of replicates

- Several masses, volumes, and temperature (for digestion)

Experimental design:

I recommend measuring the active surface area of the sorbent using BET and to carry out sorption isotherms in order to determine the maximum sorption capacity. The sorption capacity is a much more robust and general parameter to characterize the sorption efficiency of the adsorbent than the volume of solution which depends on the analyte concentration and mass of sorbent.

Results and discussion:

The advantages of the method are not demonstrated. The recovery results and LODs are not very informative if they are not compared with a control experiment (e.g. direct injection). The results of the standard addition could then be used to pinpoint matrix effects when samples are directly injected and one could compare the intensity of the matrix effect with and without preconcentration step. In addition, comparing the LODs with and without preconcentration steps and even with other standard methods used for preconcentration of transition metals. Also comparing the performances of the method with other published methods would be very important.

Standard deviation and errors’ evaluation is not provided for most of the results.